# Maternal and Neonatal Outcomes of Elective Induction of Labor at 39 or More Weeks: A Prospective, Observational Study

**DOI:** 10.3390/diagnostics13010038

**Published:** 2022-12-23

**Authors:** Soobin Lee, Dong Hyun Cha, Cho Won Park, Eui Hyeok Kim

**Affiliations:** 1CHA Bundang Medical Center, Department of Obstetrics and Gynecology, College of Medicine, CHA University, Seongnam-si 13496, Republic of Korea; 2CHA Gangnam Medical Center, Department of Obstetrics and Gynecology, CHA University School of Medicine, Seoul 06135, Republic of Korea; 3CHA Ilsan Medical Center, Department of Obstetrics and Gynecology, CHA University School of Medicine, Goyang 10414, Republic of Korea

**Keywords:** induced labor, cesarean section, obstetrical vacuum extraction, postpartum period, newborn infant

## Abstract

The purpose of our study is to compare the maternal and neonatal outcomes of induction of labor (IOL) versus expectant management at 39 weeks of gestation. We conducted a single-centered, prospective, observational study of nulliparous singleton women at 39 weeks or more. We compared the maternal and perinatal outcomes. Of 408 nulliparous women, 132 women were IOL group and 276 women were expectant management group. IOL and expectant group had similar cesarean delivery rate (18.2% vs. 15.9%, *p* = 0.570). The delivery time from admission was longer in IOL group (834 ± 527 vs. 717 ± 469 min, *p* = 0.040). The IOL group was less likely to have Apgar score at 5 min < 7 than in expectant group (0.8% vs. 5.4%, *p* = 0.023). Multivariate analysis showed that IOL at 39 weeks was not an independent risk factor for cesarean delivery (relative risk 0.64, 95% confidence interval: 0.28–1.45, *p* = 0.280). Maternal and neonatal adverse outcomes, including cesarean delivery rate, were similar to women in IOL at 39 weeks of gestation compared to expectant management in nulliparous women. IOL at 39 weeks of gestation could be recommended even when the indication of IOL is not definite.

## 1. Introduction

The question of when to electively induce labor without medical or obstetric indications has always been complex. This is because multiple factors must be considered when determining the optimum delivery time, and gestational age is an important consideration. That is, delivery before 39 weeks is avoided when possible due to the evidence that perinatal outcomes are worse in unindicated delivery before full-term than in continued pregnancy [1,2]. In addition, gestational age advancing beyond full term is known to have increased adverse maternal and neonatal complications [1,3]. Thus, the question arises as to whether inducing delivery at full term, 39 gestational weeks, is acceptable.

Induction of labor (IOL) can be advantageous for several reasons. Most importantly, unnecessary prolongation of pregnancy can be avoided, which in turn avoids unnecessary complications, such as macrosomia and its associated consequences (i.e., postpartum hemorrhage, vaginal laceration, infection, and stillbirth) and preeclampsia [4,5]. Additionally, IOL may benefit cost reduction and quality of life by scheduling one’s own labor, allowing women to prepare and plan ahead.

Despite many advantages, elective IOL in low-risk nulliparous women at full term was historically discouraged because of the long-standing belief that IOL can increase the risk of cesarean delivery and maternal/neonatal morbidities [6,7]. Especially, increasing the risk of unnecessary cesarean delivery should be avoided if possible since it could lead to difficult conditions for consecutive deliveries. For instance, repetitive cesarean sections may lead to difficult access to the lower uterine segment, induce organ damage, and increase the risk of maternal and fetal complications related to the procedure [8]. Thus, counseling was recommended by the 2009 Practice Bulletin of the American College of Obstetricians and Gynecologists (ACOG) to inform patients that elective IOL had a 2-fold increased risk of cesarean delivery [9].

In a real clinical setting, the decision should be between whether to induce labor or to expectantly manage, because not all women who are managed expectantly will spontaneously labor. Some may ultimately go through IOL while waiting for spontaneous labor or cesarean delivery without labor pain for various reasons close to the due date. Thus, a comparison between the actual delivery choices is required.

Recent studies that used a suitable comparison group did not show a higher risk of adverse outcomes with IOL. Instead, some studies even demonstrated a decreased frequency of cesarean delivery and adverse neonatal outcomes in those induced at 39 weeks compared to those expectantly managed. the randomized trial of induction versus expectant management (ARRIVE) study [10] concluded that IOL significantly decreased the risk of other maternal factors, including cesarean delivery rate and neonatal morbidities. This result led to the renewal of the previous policy statement of ACOG, which supports that non-medically indicated induction at 39 weeks for nulliparous women can be a “reasonable” option [11,12]. However, this has some limitations. First, the trial lacked diversity because it was conducted only in the US. Second, ascertainment bias was possible because the trial was randomized and unmasked. Therefore, large-scale observational studies are required in diverse settings.

In particular, evaluating the outcome of IOL in South Korea, which has the world’s lowest fertility rate (0.81 in 2021), could be of great importance [13]. Women in South Korea tend to avoid or delay getting married or having children for many social reasons [14]. This increases nulliparity and advanced maternal age, which ultimately increases high-risk pregnancies, such as diabetes and gestational hypertensive disorders [15]. Moreover, the cesarean section rate in South Korea ranks as one of the highest in the Organization for Economic Cooperation and Development’s *cesarean section rankings* [16]. Therefore, elective IOL at full term can be a useful alternative delivery method to decrease cesarean delivery rates.

The purpose of our study was to overcome these limitations and compare maternal outcomes, including cesarean delivery and perinatal outcomes, between IOL and expectant management at 39 weeks in nulliparous singleton pregnancies in Korea.

## 2. Materials and Methods

This study was a prospective observational study conducted between 1 January 2018, and 30 June 2022, at the National Health Insurance Service Ilsan Hospital in the Republic of Korea. The study was approved by the National Health Insurance Service Ilsan Hospital’s institutional review board (#NHIMC 2017-12-010), and written informed consent was obtained.

All nulliparous women at 39 gestational weeks were considered for participation in this study. We allowed the participants to choose to wait for spontaneous labor or induce labor at 39 gestational weeks. (Figure 1). The following inclusion criteria were applied: (1) nulliparous women; (2) living singleton pregnancy; (3) gestational age ≥39 weeks; and (4) vertex presentation. Patients with indications for cesarean delivery, including previous cesarean delivery and previous uterine surgery were excluded. For proper evaluation of maternal and neonatal outcomes, the following were excluded: (1) neonates with major anomalies; (2) a fetal birth weight less than 2500 g; (3) patients with a rupture of membranes before deciding whether to wait for IOL or spontaneous labor as it could alter the course of labor progress; and (4) women with active or suspected COVID-19 infection (Figure 2).

After admission to the delivery room for delivery, a pelvic examination was done for the Bishop score. Ultrasonography was performed using an EPIQ 7 (Bothell, WA, USA) with a vaginal probe for cervix length measurement with previously validated technical criteria [17]. IOL was attempted with pitocin (pitocin, intravenous injection, 10 IU/mL, Jeil Pharmaceutical Co., Ltd., Daegu, Republic of Korea) or prostaglandin E2 (Propess, intravaginal, 10 mg, Bukwang Pharm Co., Ltd., Seoul, Republic of Korea). Whether to use pitocin or prostaglandin E2 was decided based on the favorability of the cervix. Patients with an unfavorable cervix, defined as Bishop score ≤4, had prostaglandin E2 vaginally inserted [9]. It was removed after 24 h of insertion or earlier in case of onset of active labor, rupture of membranes, or abnormal cardiotocography. If patients had a favorable cervix (Bishop score >4) or inadequate contraction for labor progress, pitocin was used.

All maternal and neonatal demographics were obtained from the hospital’s electronic medical records. Maternal demographics were as follows: maternal age, height, weight before pregnancy and delivery, with which we calculated the body mass index (BMI), gestational weeks at delivery, and cervical status (i.e., Bishop score and cervical length before admission for delivery). To compare the adverse maternal outcomes between the two groups, we evaluated several variables, including the cesarean delivery rate, the time between delivery and admission of IOL (duration between admission to delivery room and delivery time) and admission in the expectant management group, duration of the second stage (when the cervix is fully dilated and the baby passes through the birth canal until delivery), decrement in hemoglobin, postpartum uterine embolization due to massive bleeding, transfusion rate, length of hospital stay, readmission rate within 30 days, and revisit within 50 days after discharge [18]. Data were obtained from the institutional electronic medical records.

Considering the puerperium during which the mother’s reproductive system returns to its normal pre-pregnant state and generally lasts for six subsequent weeks postpartum, cesarean section patients usually have a six-week postpartum visit and most post-cesarean complications occur during the puerperium. We chose this timeline of 50 days after discharge with reference to a previous study [18]. We used readmission rates within 30 days post-discharge because most perioperative mortality and morbidity occur during this period, according to a previous study [19]. To compare adverse neonatal outcomes, we investigated fetal birth weight, 5-min Apgar score, neonatal intensive care unit admission rate, meconium status, and intubation status.

For statistical analyses, demographic and clinical characteristics were compared between patients with IOL and expectant management using Student’s *t*-test for continuous values and the χ2 test or Fisher’s exact test for categorical values. Multivariate analysis using a logistic regression model was performed to determine whether IOL could be an independent risk factor for cesarean delivery. All *p*-values were 2-tailed, and *p* < 0.05 was considered statistically significant. All analyses were performed using Statistical Package for Social Sciences version 23.0 (SPSS Inc., Chicago, IL, USA).

## 3. Results

A total of 536 nulliparous women at ≥39 weeks of gestation who consented to participation were recruited for this study. After excluding 5 cases with fetal anomalies, 32 with a maternal request for cesarean delivery before labor, 20 with breech presentation, 36 with or suspected COVID-19 infection, 32 with neonatal weight less than 2500 g, and one woman who experienced fetal death, the remaining 408 patients were included in the final analysis. Of these, two groups were divided and compared: the IOL group with 132 women (32.4%) and the expectant group with 276 women (67.6%) (Figure 2).

The demographic characteristics and clinical outcomes are shown in Table 1. The average maternal age was older in the IOL group but was not statistically significant. The gestational age at birth in the expectant management group was 39.6 ± 0.5 weeks. BMI before pregnancy (22.0 ± 3.9 vs. 21.1 ± 2.7, *p* = 0.004) and at term (27.1 ± 4.2 vs. 26.2 ± 3.3, *p* = 0.021) were significantly higher in the expectant management group. However, overall weight gain during pregnancy was similar between the two groups. The average Bishop score at admission was lower in the IOL group, which means a more unfavorable cervix (3.8 ± 1.6 vs. 4.1 ± 1.8, *p* = 0.170) and longer cervix length in the IOL group, although this was not statistically significant (20.1 ± 0.9 vs. 18.6 ± 8.6, *p* = 0.132)

A comparison of the maternal outcomes between the two groups is shown in Table 2. The cesarean delivery rate was similar between the IOL and expectant groups (18.2% vs. 15.9%, *p* = 0.570). The rate of operative vaginal delivery with vacuum was similar between the two groups. The delivery time from intervention or admission was significantly longer in the IOL group than in the expectant management group (834 ± 527 vs. 717 ± 469 min, *p* = 0.040). However, the rate of delivery within 12 h and the duration of the second stage of labor were similar between the two groups. The length of hospital stay did not differ between the two groups. Regarding maternal complications, hemoglobin decrement during delivery, transfusion rate, and uterine artery embolization when massive postpartum bleeding occurred, did not differ between the two groups. Other maternal outcomes such as readmission rate within 30 days and outpatient visits more than 2 times within 50 days were similar between the two groups.

The neonatal outcome of the two groups are compared in Table 3. According to our analysis, neonatal weight was slightly higher in the expectant management group, but the difference was not statistically significant. Concerning neonatal complications, 1 min and 5 min Apgar scores were similar between the groups; however, the rate of 5 min Apgar scores <7 was higher in the expectant management group (0.8% vs. 5.4%, *p* = 0.023). The rates of intubation and meconium-stained amniotic fluid were similar between the IOL and expectant management groups. Nevertheless, the expectant management group had a statistically significant higher risk of NICU admission rate than the IOL group (49.2% vs. 36.6%, *p* = 0.015).

The results of the univariate logistic regression analysis are presented in Table 4. In pregnant women at ≥39 weeks of gestation, IOL was not found to be an independent predictor of cesarean delivery (relative risk [RR] 0.83, 95% confidence interval (CI): 0.48–1.44, *p* = 0.507).

Further study was done by multivariate analysis, where we adjusted factors that could affect the delivery outcome, such as maternal age, gestational age, neonatal birthweight larger than 3500 g, and maternal pre-pregnancy BMI. Our analysis concluded that IOL was still not considered an independent risk factor for cesarean delivery (RR = 0.64; 95% CI: 0.28–1.45, *p* = 0.280), as shown in Table 4.

We have summarized the main findings of our analysis in terms for pregnancy-maternal and fetal outcomes in Table 5.

## 4. Discussion

Through this prospective observational study involving uncomplicated nulliparous women, we found that elective IOL at 39 weeks was not related to an increase in the number of cesarean deliveries or worsened perinatal and neonatal outcomes compared to the expectant management group. In particular, with a longer cervix and lower Bishop score at admission, IOL was not an independent predictor of cesarean delivery rate. Moreover, more than 80% of nulliparous women at ≥39 weeks of gestation delivered vaginally, irrespective of the group.

These findings are similar to those of recent, large, randomized trials. For example, Saccone et al. [20] concluded that IOL at 39 weeks does not increase the risk of cesarean delivery compared to the expectant management group. This is consistent with our study and proves that the previous proposition that IOL increases the cesarean delivery rate is inaccurate. On the other hand, some studies presented a lower cesarean delivery rate in low-risk women undergoing IOL at 39 weeks than in those who were expectantly managed [10]. This difference may be explained by several unfavorable characteristics in the IOL group for successful IOL, such as older age, higher BMI, longer cervical length, and lower Bishop scores [21].

Additionally, despite the previous belief in poorer maternal outcomes, our study showed that there were no statistically significant differences in operative vaginal delivery rate, transfusion rate, embolization, readmission rate after discharge, and outpatient visits after discharge between the IOL and expectant groups, indicating that IOL is not related to adverse outcomes, which is concurrent with other recent studies [10,20,22]. Moreover, our study demonstrated an increased admission-to-delivery time in the IOL group just as in the ARRIVE trial and the recent retrospective cohort study performed by Souter et al. [23]. Hence, both studies showed concerns of increased patient costs. However, admission-to-delivery time did not increase the total length of hospital stay. This suggests that no more hospital resources were used in the IOL group than in the expectant group. Furthermore, in order to analyze the economic cost of IOL in detail, the resources used and costs in the antepartum and postpartum periods should also be considered. Since postpartum outpatient visits and readmission rates showed no significant difference between the two groups, it may be difficult to conclude with increased costs. Thus, the economic impact of IOL requires further evaluation.

Regarding neonatal outcomes, our study presented a lower frequency of 5-min Apgar scores <7 and a higher NICU admission rate in the IOL group than in the expectantly managed group. The higher NICU admission rate was presumed to be due to the early amniotomy for induction. Artificial amniotomy to induce labor could cause prolonged rupture of membrane, and when the duration is more than 18 h before delivery, the pediatric department hospitalizes neonates to the NICU for close observation of neonatal sepsis. Although previous observational studies vary in the neonatal outcomes of IOL, it ultimately arrives at the same conclusion that elective induction does not worsen neonatal outcomes.

This study has several strengths. First and most importantly, this is a carefully designed prospective observational study, which allowed us to minimize recall bias and estimate the real risks of IOL more efficiently. Second, we collected complete patient records from one institution, using a uniform protocol for analysis. With these data, multivariate analyses could be performed with the correct variables to consider potential confounding factors for severe maternal outcomes, such as BMI or accurate gestational weeks, and all participants were followed up until postpartum outpatient department visits, providing more detail. Third, women were carefully selected so only low risk, nulliparous women were included, and women with premature rupture of membranes were excluded because it may alter labor progress, unlike other indications for IOL. Finally, to our knowledge, our study is the first prospective observational study conducted in east Asia. Although China recently reported a nationwide cross-sectional study of IOL and its outcomes, the comparator group was not suitable, resulting in inadequate results [24].

This study has some limitations. First, this study was conducted with a relatively small number of women from one institution, which may limit the generalizability of the findings. Furthermore, because most maternal and perinatal complications, except cesarean delivery and NICU admission rate, were relatively uncommon, this study could not detect differences in critical complications. Second, masking was difficult, and the potential for confirmation bias is possible. Finally, as this study was tried in one single center with a uniform protocol for IOL, the results could not be applied to other hospitals.

## 5. Conclusions

In conclusion, this study showed that elective IOL in uncomplicated nulliparous women at 39 weeks is not related to an increase in cesarean delivery or worse maternal and neonatal outcomes compared to expectant management despite an unfavorable cervix. This result suggests that IOL at 39 weeks could be considered for the purpose of controlling one’s time of birth without any definite indication of induction. However, proper education and tailored counseling of nulliparous women at 39 or more gestational weeks is crucial for making an informed decision. Moreover, further research on a larger scale is necessary to accurately assess the obstetric and neonatal effects of IOL at 39 weeks.

## Figures and Tables

**Figure 1 diagnostics-13-00038-f001:**
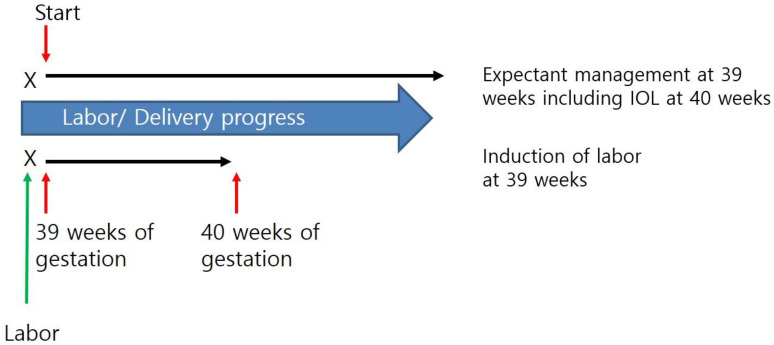
Diagram of study design. IOL, induction of labor.

**Figure 2 diagnostics-13-00038-f002:**
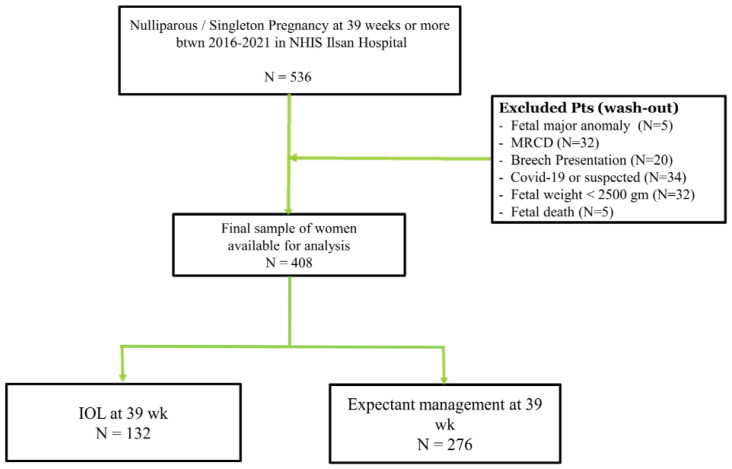
Study flow chart. IOL, induction of labor; MRCD, maternal request cesarean delivery; NHIS, National Health Insurance Service; Pts, patients.

**Table 1 diagnostics-13-00038-t001:** Characteristics of women undergoing induced labor and expectant management.

Characteristics	Induction of Labor (*n* = 132)	Expectant Management (*n* = 276)	*p*-Value
Age (year)	32.6 ± 4.7	31.7 ± 4.2	0.060
Gestational age (weeks)	39.0	39.6 ± 0.5	<0.001 *
Weight gain in pregnancy	12.1 ± 6.0	12.9 ± 5.5	0.194
BMI			
BMI in pre-pregnancy	22.0 ± 3.9	21.1 ± 2.7	0.004 *
BMI in term	27.1 ± 4.2	26.2 ± 3.3	0.021 *
Bishop Score ** at admission	3.8 ± 1.6	4.1 ± 1.8	0.170
Cervix length (mm) at admission	20.1 ± 0.88	18.6 ± 8.6	0.132

Data are presented as the median (range), mean ± standard deviation. BMI, body mass index. * Statistical significance, ** Total possible score = 13.

**Table 2 diagnostics-13-00038-t002:** Obstetrics and maternal outcomes of induced labor and expectant management.

Variables	Induction of Labor (*n* = 132)	Expectant Management (*n* = 276)	*p*-Value
Cesarean section	24 (18.2)	44 (15.9)	0.570
Operative vaginal delivery	21 (18.9)	41 (17.2)	0.700
Length of hospital stay (day)	4.1 ± 1.3	4.0 ± 1.3	0.244
Time from admission to delivery (min)	835 ± 527	717 ± 469	0.040 *
Time for second stage labor (min)	69 ± 54	76 ± 51	0.276
Delivery within 12 h	58 (53.7)	142 (61.2)	0.191
Decrease in Hgb after delivery (g/dL)	1.7 ± 2.1	1.8 ± 1.7	0.526
Transfusion	3 (2.3)	13 (4.7)	0.235
Embolization	2 (1.5)	1 (0.4)	0.202
Readmission **	4 (3.0)	17 (6.2)	0.181
Outpatient visits *** > 2	73 (55.3)	154 (55.8)	0.925

Data are presented as the number (%) or the median (range), mean ± standard deviation. Hgb, hemoglobin. * Statistical significance, ** within 30 days of discharge, *** within 50 days of discharge.

**Table 3 diagnostics-13-00038-t003:** Neonatal outcomes of induced labor and expectant management.

Variables	Induction of Labor (*n* = 132)	Expectant Management (*n* = 276)	*p*-Value
Fetal body weight (gm)	3233 ± 376	3304 ± 341	0.057
FBW > 3500 gm	34 (25.8)	75 (27.2)	0.762
Apgar score			
AS at 5 min	8.2 ± 1.1	8.1 ± 1.4	0.421
AS at 5 min < 7	1 (0.8)	15 (5.4)	0.023 *
NICU admission	65 (49.2)	101 (36.6)	0.015 *
Intubation	3 (2.3)	9 (3.3)	0.581
Meconium-stained amniotic fluid	18 (13.6)	51 (18.5)	0.222

Data are presented as the number (%) or the median (range), mean ± standard deviation. FBW, fetal body weight; AS, Apgar score; NICU, neonatal intensive care unit. * Statistical significance.

**Table 4 diagnostics-13-00038-t004:** Logistic regression for successful vaginal delivery.

	Odds Ratio (95% CI)			
	Unadjusted OR	*p*-Value	Adjusted OR	*p*-Value
Induction of labor at 39 weeks	0.83 (0.48–1.44)	0.507	0.64 (0.28–1.45)	0.280

OR, odds ratio; CI, confidence interval; Adjusted for maternal age, pre-pregnancy BMI, gestational age at birth, and neonatal weight larger than 3500 gm.

**Table 5 diagnostics-13-00038-t005:** Summary of obstetrics and maternal-fetal outcomes of induced labor and expectant management.

	Induction of Labor	Expectant Management
**Maternal outcome**		
Cesarean section	Similar	Similar
Time from admission to delivery	Shorter	Longer
Time for second stage labor	Similar	Similar
Postpartum hemorrhage	Similar	Similar
Readmission *	Similar	Similar
**Neonatal outcome**		
AS at 5 min <7	Lower	Higher
NICU admission	Higher	Lower
Intubation	Similar	Similar
Meconium-stained amniotic fluid	Similar	Similar

Data are presented as the number (%) or the median (range), mean ± standard deviation. AS, Apgar score; NICU, neonatal intensive care unit. * Statistical significance, * within 30 days of discharge.

## Data Availability

Data is unavailable due to privacy.

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
