# Peer review of "Maternal and Neonatal Outcomes of Elective Induction of Labor at 39 or More Weeks: A Prospective, Observational Study"

_diagnostics, 2022, doi:10.3390/diagnostics13010038_

Round 1

Reviewer 1 Report

The authors conducted a prospective observational study to determine perinatal and neonatal outcomes of elective-induced labor at 39 or more gestational weeks. This is a well-designed single-center study, which provides valuable insights regarding elective labor induction outcomes in low-risk, nulliparous women. The introduction provides sufficient background information, the methods and results are presented clearly, and the discussion provides enough questions to be addressed in future studies. The references are adequate, relevant, and up-to-date, and the English language needs minor spell-checking.

I have a few additional specific questions and comments for the authors:

1. Introduction, lines 30-34: "Although the primary goal is to optimize maternal and perinatal health, the best decision may differ for pregnant women and babies. This is a challenge for many obstetrical medical staff because multiple factors must be considered when determining the optimum delivery time and method. However, there is a general 33 agreement that gestational age is an important consideration."  My suggestion is to remove the first sentence. Moreover, merging the following sentences would improve the grammar and scientific soundness of the paragraph in question.

2. Introduction, lines 52 and 53: "However, this misguided recommendation is based on studies with inappropriate comparison groups, which compared IOL with spontaneous labor." The authors should remove these sentences for the scientific purposes of the paragraph. 

3. Materials and Methods, lines 89 and 90: "We allowed the participants to choose to wait for spontaneous labor or induce labor at 39 gestational weeks." The participant's reasons for choosing the labor induction or waiting for the spontaneous labor should provide enough material for a separate study, and I encourage the authors to pursue this subject.   

4. Materials and Methods lines 107 and 108: You should state on what basis You chose to induce labor with Pitocin or prostaglandin E2. 

5. Materials and Methods line 119: You should state the meaning of "time between delivery and intervention of IOL"

6. Materials and Methods line 126 and 127: "...most postoperative complications occur within 50 days". This statement requires a reference.  

7. References 10, 12, and 15: You left "accessed on..." in these references. 

Author Response

Dear Editor

We are very pleased to have the opportunity to revise our manuscript for ‘Diagnostics’. We deeply appreciate your comments and those of the reviewers, which were very helpful in improving our paper. We have carefully considered them and revised our manuscript accordingly.

We have tried our best to address all the comments. Our revisions are described as follows. The corresponding changes in the manuscript are indicated in red font.

Comments from the Editors and Reviewers:

Reviewer 1>

Question 1. Introduction, lines 30-34: "Although the primary goal is to optimize maternal and perinatal health, the best decision may differ for pregnant women and babies. This is a challenge for many obstetrical medical staff because multiple factors must be considered when determining the optimum delivery time and method. However, there is a general 33 agreement that gestational age is an important consideration."  My suggestion is to remove the first sentence. Moreover, merging the following sentences would improve the grammar and scientific soundness of the paragraph in question.

Answer 1. Thanks for your comment, we removed this first sentence as you advised and merged to simpler sentence. (page1, line 30-31)

Question 2. Introduction, lines 52 and 53: "However, this misguided recommendation is based on studies with inappropriate comparison groups, which compared IOL with spontaneous labor." The authors should remove these sentences for the scientific purposes of the paragraph.

Answer 2. Thank you for comment. We removed these sentences according to your suggestion

Question 3. Materials and Methods, lines 89 and 90: "We allowed the participants to choose to wait for spontaneous labor or induce labor at 39 gestational weeks." The participant's reasons for choosing the labor induction or waiting for the spontaneous labor should provide enough material for a separate study, and I encourage the authors to pursue this subject.

Answer 3. Thank you for the comment. We agree that the reasons for choosing the labor induction or waiting for the spontaneous labor should be further studied. After further study we will reflect this on our next report.

Question 4. Materials and Methods lines 107 and 108: You should state on what basis You chose to induce labor with Pitocin or prostaglandin E2.

Answer 4. Thanks for your comments.

If the cervix was unfavorable with bishop score 4 or less, we used prostaglandin 2 as ripening agent and if the cervix was favorable, we used the Pitocin. We added this contents with reference in the manuscript. (Page3-4, Line 109-114)

Whether to use pitocin or prostaglandin E2 was decided based on the favorability of the cervix. Patients with unfavorable cervix, which is defined as Bishop score ≤4, had prostaglandin E2 vaginally inserted.[9] It was removed after 24 hours of insertion or earlier in case of the onset of active labor, rupture of membranes, or abnormal cardiotocography. If patients with favorable cervix (Bishop score >4) or inadequate labor for labor progress, pitocin was used.

Question 5. Materials and Methods line 119: You should state the meaning of "time between delivery and intervention of IOL"

Answer 5. We have added further information. Thank you. (Page4, Line 121-122)

Duration between admission to delivery room and delivery time

Question 6. Materials and Methods line 126 and 127: "...most postoperative complications occur within 50 days". This statement requires a reference. 

Answer 6. Thank you for comments

Puerperium during which the mother’s reproductive system returns to its normal prepregnant state, generally lasts six subsequent weeks postpartum and most post-operation complications occur during the puerperium.  

We have added the contents with the reference. (Page4, Line 126)

Considering the puerperium during which the mother’s reproductive system returns to its normal pre-pregnant state and generally lasts six subsequent weeks postpartum , cesarean section patients usually have a six-week postpartum visit and most post-cesarean complications occur during the puerperium, we have chosen this timeline of 50 days after discharge with reference to previous study.  

Question 7. References 10, 12, and 15: You left "accessed on..." in these references.

Answer 7. Thank you. We have added the accessed date on each reference. (Reference 11, 13, 16)

Thank you very much 

The final draft is in the attachment file. 

Reviewer 2 Report

Dear authors, it has been a pleasure to read your paper, whether to induce or not labor universally is going to be a life long debate, but paper like yours sharing experience regarding big sample of patients is of crucial importance to allow future decision and consideration within guidelines, therefore I absolutely support the topic

the paper is well written, the analysis well conducted

i have just minor revisions to suggest

1) please mention within the introduction or the discussion the crucial role of preventing first Caesarean section due to the risks associated with difficult further CS, I suggest to read and cite PMID: 31962259.

2) I would like you do realize a table with a summary of the main findings with your analysis in terms of pregnancy-maternal fetal outcome to help the reader with an immediate understanding of your results

otherwise well done

Author Response

Dear Editor

We are very pleased to have the opportunity to revise our manuscript for ‘Diagnostics’. We deeply appreciate your comments and those of the reviewers, which were very helpful in improving our paper. We have carefully considered them and revised our manuscript accordingly.

We have tried our best to address all the comments. Our revisions are described as follows. The corresponding changes in the manuscript are indicated in red font.

Comments from the Editors and Reviewers:

Question 1. Introduction: Please mention within the introduction or the discussion the crucial role of preventing first Caesarean section due to the risks associated with difficult further CS, I suggest to read and cite PMID: 31962259.

Answer 1. We have referred to the material you kindly gave us. We added the following statement on introduction and explained the importance of preventing first cesarean section. Thank you. (Page2, Line 46-50).

Especially, increasing the risk of unnecessary cesarean delivery should be avoided if possible since it could lead to difficult condition for consecutive delivery. For instance, repetitive cesarean section may lead to difficult access to the lower uterine segment, induce organ damage and increase risk of maternal and fetal complications related to the procedure increases

Question 2. Table : I would like you do realize a table with a summary of the main findings with your analysis in terms of pregnancy-maternal fetal outcome to help the reader with an immediate understanding of your results.

Answer 2. I have included a new table for summary or our main findings. Table 5 in Results.

Thank you very much

The final draft is in the attachment file